# Mechanical Improvement of Graphene Oxide Film via the Synergy of Intercalating Highly Oxidized Graphene Oxide and Borate Bridging

**DOI:** 10.3390/nano15080630

**Published:** 2025-04-20

**Authors:** Yiwei Quan, Peng He, Guqiao Ding

**Affiliations:** 1National Key Laboratory of Materials for Integrated Circuits, Shanghai Institute of Microsystem and Information Technology, Chinese Academy of Sciences, 865 Changning Road, Shanghai 200050, China; qyw15028595035@163.com; 2College of Materials Science and Opto-Electronic Technology, University of Chinese Academy of Sciences, Beijing 100049, China

**Keywords:** graphene oxide films, orientation, mechanical properties, hydrogen bonding, covalent crosslinking

## Abstract

Converting graphene oxide (GO) nanosheets into high-performance paper-like GO films has significant practical value. However, it is still challenging because the mechanical properties significantly decreased when the nanosheets are assembled into films. The simultaneous attainment of high tensile strength, high modulus, and relatively high toughness remains a formidable challenge. Here, we demonstrated an effective approach involving the incorporation of high oxidized graphene oxide (HOGO) and borate, to enhance the mechanical properties of GO films. X-ray photoelectron spectroscopy (XPS) measurements and thermogravimetric analysis-differential scanning calorimetry (TG-DSC) revealed the synergistic effects of hydrogen and covalent bonding from HOGO and borate, respectively. Additionally, wide-angle X-ray scattering (WAXS) analysis indicated a notable enhancement in the orientation of the GO in the resulting films, characterized by the Herman’s orientation factor (ƒ = 0.927), attributable to the combined action of hydrogen and covalent bonding. The borate-crosslinked GO+HOGO films exhibited exceptional mechanical properties, with an impressive strength (417.2 MPa), high modulus (43.8 GPa), and relatively high toughness (2.5 MJ m^−3^). This innovative assembly strategy presents a promising avenue for achieving desirable mechanical properties, thereby enhancing the potential for commercial applications.

## 1. Introduction

As the water-soluble derivative of graphene, graphene oxide (GO), with many functional groups on the surface, is one of the best candidates for fabricating GO films with high mechanical properties, because oxygen-containing functional surface groups allow for chemical cross-linking to improve the interfacial strength of the adjacent GO layers [1]. GO films with high mechanical properties have found widespread applications in various commercial fields, including photovoltaic devices [2,3], nanogenerators [4], and mechanical engineering [5,6], which need the construction of chemical bonding links. However, pure GO films exhibit low mechanical properties owing to a weak interlayer connection. Drawing inspiration from a plethora of natural materials, such as bone [7], nacre [8], wood [9], bamboo [9], and others [10,11], some research has been undertaken to enhance the mechanical properties of GO films [5,12,13], particularly focusing on modulus improvements [5,14,15,16]. Park et al. demonstrated a method wherein carboxylate groups on the edges were bonded with Mg^2+^ and Ca^2+^ ions, yielding high modulus films with values of 27.9 ± 1.8 GPa for Mg-modified films and 28.1 ± 1.2 GPa for Ca-modified films [12]. However, the tensile strength of these films was relatively low, measuring only 80.6 ± 16.5 MPa for Mg-modified films and 125.8 ± 13.6 MPa for Ca-modified films. An et al. introduced borate crosslinking into the interlayer of GO membranes, resulting in an ultra-high modulus (127.0 ± 4.0 GPa) but compromised tensile strength (185.0 ± 30.0 MPa) and toughness (0.1 MJ m^−3^) [5]. Therefore, although the incorporation of various small molecules and ions facilitated interconnections between GO functional groups, thereby enhancing modulus, it has often failed to improve strength and toughness [17,18,19]. GO films with a high modulus but low strength and toughness are deemed unsuitable for commercial applications.

Several studies have concentrated on synergistically enhancing the mechanical properties of GO membranes [20,21,22], particularly focusing on both strength and toughness [1,13,23,24]. Cheng et al. introduced 10,12-pentacosadiyn-1-ol (PCDO) into the interlayer of GO films, followed by reduction with HI, resulting in impressive tensile strength and toughness values of 156.8 MPa and 3.9 ± 0.03 MJ m^−3^, respectively, by π–π interactions and covalent bonding [1]. Zhang et al. harnessed π–π interactions in addition to hydrogen bonding between adjacent GO nanosheets through phenylboronic acid moieties, achieving remarkable tensile strength and toughness of 382.0 MPa and 7.5 MJ m^−3^, respectively [13]. More recently, Gong et al. implemented GO-based nanocomposite films via a gel-film transformation method. Synergistic enhancement was achieved by hydrogen and ionic bonding with carboxymethyl cellulose (CMC) and intercalating Mn^2+^ ions in the GO-based nanocomposite films, resulting in exceptionally high strength (526.7 MPa) and high toughness (6.6 MJ m^−3^), while the modulus remained at 12.6 GPa [25]. Nonetheless, the simultaneous attainment of high strength, high modulus, and relatively high toughness remains a formidable challenge for GO films [20].

Constructing synergy bonds provides a workable approach for enhancing the mechanical properties of GO films. In this study, we aimed to enhance the mechanical properties of GO films through a synergistic approach involving intercalating high oxidized graphene oxide (HOGO) and borate bridging. Our investigation, employing X-ray photoelectron spectroscopy (XPS) measurements and thermogravimetric analysis differential scanning calorimetry (TG-DSC), revealed the synergistic effects of hydrogen and covalent bonding. Additionally, wide-angle X-ray scattering (WAXS) analysis indicated a notable enhancement in the orientation of the resulting GO films, characterized by the Herman’s orientation factor (ƒ = 0.927), attributable to the combined action of hydrogen and covalent bonding. The borate-crosslinked GO+HOGO films exhibited exceptional mechanical properties of an impressive strength (417.2 MPa), high modulus (43.8 GPa), and a relatively high toughness (2.5 MJ m^−3^). Importantly, these values significantly surpassed those of both the individual GO and HOGO films, with improvements of 1.6-, 2.2-, 3.0- and 2.3-, 3.6-, 5.2-fold compared to GO+B and HOGO+B films in our work, respectively. This innovative strategy presents a promising avenue for achieving high strength and modulus coupled with other desirable mechanical properties, thereby enhancing the potential for commercial applications.

## 2. Materials and Methods

### 2.1. Materials

Deionized water with a resistivity of 18.3 MΩ·cm^−1^ was obtained from a QH-DI-15 water purification system purchased from Qianhe Water Treatment Engineering Co., Ltd., Dongguan, China. Graphite flakes (30 mesh and 300 mesh) were sourced from Yunshen Graphite Co., Ltd., Qingdao, China. Chemicals including sulfuric acid (H_2_SO_4_ ≥ 98%), potassium permanganate (KMnO_4_ ≥ 99.5%), hydrogen chloride (HCl, 36.0−38.0%), ethanol (≥98.0%), acetone (≥99.5%), hydrogen peroxide (H_2_O_2_ ≥ 30.0%), sodium tetraborate (Na_2_B_4_O_7_·10H_2_O ≥ 99.0%) and ammonium hydroxide (NH_4_·OH, 25.0–28.0%), were purchased from Sinopharm Chemical Reagent Co., Ltd., Shanghai, China. Polytetrafluoroethylene (PTFE) membranes with a pore diameter of 0.22 μm were purchased from Yibo Filtration Equipment Co., Ltd., Haining, China. The polyethylene terephthalate (PET) substrate was sourced from Dongxuan Plastic Products Co., Ltd., Suzhou, China. A silicon wafer was purchased from the Photovoltaic Laser Institute, Chinese Academy of Sciences, China. All reagents were used as received without further purification.

### 2.2. Preparation of GO Nanosheets

The method used for preparing GO nanosheets was adapted from our previous work, using electrochemical delamination [26]. Initially, 10 g of graphite flakes (approximately 30 mesh) were dispersed in a 5000 mL electrolyte solution of H_2_SO_4_ using continuous magnetic stirring (300 rpm). Two Pt foils (10 cm × 3 cm) served as electrodes, positioned 2 cm apart (with 3 cm submerged), while a constant current of 5 A was applied to facilitate the electrochemical delamination of graphite flakes, with temperature control via water circulation at 25 °C. The resulting porous graphene networks (PGNs) were utilized as raw material for subsequent oxidation. To initiate oxidation, the excess H_2_SO_4_ in the electrochemical delamination process was removed, followed by the addition of 10 g KMnO_4_ (1:1 ratio) to the mixture of PGNs and H_2_SO_4_ at 5 °C. Then, the temperature was raised to 35 °C, and the agitation was terminated to enable the agitation-free oxidation process. After 40 min of oxidation, the product was filtered using a 100-mesh filter to eliminate any remaining oxidant. The filtered product was then immersed in deionized water at 0 °C, and H_2_O_2_ was added dropwise to consume excess Mn^7+^. Following the addition of sufficient HCl, the suspension was filtrated to obtain porous graphene network oxide (PGNO). Afterward, the resulting PGNO was poured into enough deionized water (0 °C) and washed via spontaneous sedimentation until neutral. The neutral PGNO was centrifugated at different centrifugal rate (3000, 6000, 9000, 12,000, and 15,000 r min^−1^, 15 min each) with the ice added to maintain a temperature of 0 °C. Then, after the final centrifugation step (15 min, 15,000 r min^−1^), the mixture was diluted in 0 °C water to achieve a concentration of approximately 1 mg/mL. Finally, the solution was put into a shaker for powerful oscillation for 3 h in order to obtain a single layer of GO. The average size of GO nanosheets (64.28 μm) was determined through field-emission scanning electron microscopy (FESEM), as depicted in Figure 1a,b. Further details on GO preparation can be seed in reference [26].

### 2.3. Preparation of HOGO Nanosheets

The method used for preparing the HOGO nanosheets was a modified version of Hummer’s method. First, 30 g of graphite flakes (approximately 300 mesh) was dispersed into 3.5 L H_2_SO_4_ with continuous magnetic stirring (at 0 °C, 200 rpm). Subsequently, while maintaining the temperature at 0 °C, 120 g KMnO_4_ (1:4 ratio) was added to the mixture (at 300 rpm). The temperature of the system was then gradually increased to 35 °C, where it remained for 10 h. Centrifugal cleaning and powerful oscillation steps were employed to obtain HOGO, referring to the process described above (in Section 2.2). The average size of the obtained HOGO nanosheets (3.51 μm) can be observed in Figure 1c,d.

### 2.4. Preparation of Borate-Crosslinked GO+HOGO Films

The as-prepared GO solution (1 mg/mL) was concentrated into a GO slurry (20 mg/mL) through centrifugation at 15,000 rpm for 20 min. The proposed fabrication process for borate-crosslinked GO+HOGO films, enhanced by the synergistic strengthening of hydrogen and covalent bonding, is illustrated in Figure 2a. HOGO nanosheets obtained in this process (20 mg/mL) were introduced into the prepared GO slurry and stirred at 800 rpm for 24 h (GO+HOGO = 35 g). The mass fractions of HOGO in GO were 2.0, 4.0, 8.0, and 16.0 wt %, respectively (defined as GO+2HOGO, for example). Subsequently, sodium tetraborate (0.1 mol L^−1^, 2, 4, 8, and 16 mL, equals 0.03, 0.06, 0.12, and 0.24 wt %, respectively) was added to the GO+HOGO mixture (defined as GO+4HOGO+0.03B, for example) and stirred at 800 rpm for 48 h. The resulting GO+HOGO/borate slurry was subjected to centrifugal defoaming and then poured onto the PTFE/PET substrate. The doctor blade casting was employed to orient the slurry on the substrate. The sample was air-dried at 50 °C for 12 h, followed by heating at 85 °C for 4 h, to induce crosslinking, thereby forming the borate-crosslinked GO+HOGO film. For comparative analysis, samples designated as GO, HOGO, and GO+HOGO films underwent identical processing steps, with the exception of the adding of borate to the mixture and the subsequent 85 °C crosslinking process.

### 2.5. Characterization

FESEM images were captured using the ZEISS Sigma 300 (Jena, Germany) operating at an acceleration voltage of 5 kV. XPS measurements were performed on a Thermo ESCALAB 250XI (Waltham, MA, USA) equipped with a monochromatic Al-Kα X-ray source. TG-DSC (STA449F5, NETZSCH, Selb, Germany) was conducted under a nitrogen atmosphere from room temperature to 800 °C using a temperature scan rate of 10 °C/min. Wide-angle X-ray scattering (WAXS) measurements were conducted on a Nano-star SAXS System (Billerica, MA, USA) using an incident Cu-Kα x-ray beam that was parallel to the sheet plane. The distances between the sample and detector for WAXS were 60 mm and the samples for WAXS measurements were rectangle strips measuring 1 mm wide and 10 mm long, prepared through laser cutting with a laser emitter (FLP-F50, Zörbig, Germany). The scattering patterns were collected by a Vantec (Billerica, MA, USA) 2000 detector. The degree of alignment of the graphene platelets was quantified using Herman’s orientation factor (ƒ), which is defined as Equation (1):(1)$$ƒ=\frac{1}{2}\left(3\langle \cos^{2}\phi \rangle -1\right)$$
where $\langle \cos^{2}\phi \rangle$ is the average value of the square of the cosine of the azimuthal angle ϕ for the 001 peak of GO sheets and the 002 peak of graphene sheets, which was calculated using Equation (2), as follows:(2)$$\langle \cos^{2}\phi \rangle =\frac{\int_{0}^{\pi /2}I(\phi )\cos^{2}\phi \sin\phi d\phi }{\int_{0}^{\pi /2}I(\phi )\sin\phi d\phi }$$
where I(ϕ) is the intensity at an azimuthal angle of ϕ. The intensity of the WAXS distribution in the frequency domain images of GO papers demonstrated a good fit using the Gauss distribution, according to Equation (3):(3)$$y=y_{0}+A*\exp\left(-\frac{1}{2}{\left(\frac{x-x_{c}}{w}\right)}^{2}\right)$$
where w is the full width at half-maximum (FWHM), presenting the distribution of the azimuthal angle, which indicates the alignment of the GO nanosheets. $x_{c}$ is the distribution representing the azimuthal angle corresponding to the peak of the curve. Tensile stress strain curves were obtained at a loading rate of 0.1 mm/min (using a CTM 8030 system with a 100 N load cell, Shanghai, China) under 45% humidity and a temperature of 25 °C. Samples for tensile testing were first placed on a laser cutter using a laser emitter, with 5 W power and a cutting speed of 10 mm/s. Then, the samples were cut off evenly with a scalpel to prevent the uneven surfaces from causing testing errors. The length, width, and thickness of the samples were 30, 2 mm, and 10 ± 2 μm, respectively. The mechanical properties for each sample were measured under the same condition, excluding the few sample strips that broke near the clamps from the calculations.

The tensile strength and strain of the GO films were obtained from the tensile testing data described above by the CTM 8030 system. The modulus of the GO films was obtained by differentiating the stress-strain curve [27]. The toughness of the GO films was obtained by calculating the definite integral of the stress strain curve.

## 3. Results and Discussion

### 3.1. Characterization of the Synergistic Enhancement

To illustrate the reaction more clearly, Fourier transform infrared (FT-IR) spectroscopy was used. To our knowledge, borate can be bridged to GO nanosheets by borate ortho-esters (B–O–C) with hydroxyl groups at ambient atmosphere [5,28]. As shown in Figure 2b, the FT-IR measurements show the hydroxyl groups of the GO nanosheets. When the HOGO nanosheets were added to the GO solution, as shown in Figure 2c, the hydroxyl groups of GO+HOGO film increased a little because of the abundant hydroxyl groups on the surface of the HOGO nanosheets. As shown in Figure 2d, the peak of the hydroxyl group disappeared when borate was added to the GO+HOGO films, which indicates the formation of B–O–C ortho-esters [5]. The defects and the layer distance of the GO-based films were reduced by the HOGO nanosheets and borate, as shown in Appendix A. As shown in Appendix A, the D/G ratio of the GO, GO+HOGO, GO+HOGO+B films were 0.99, 0.98 and 0.88, respectively, indicating a progressive reduction in intrinsic defects within the film structure. X-ray diffraction (XRD) analysis revealed a gradual decrease in interlayer spacing with the addition of both borate and HOGO, as shown in Appendix A.

To validate the synergistic enhancement between hydrogen and covalent bonding of our borate-crosslinked GO+HOGO films, XPS and TG-DSC measurements were performed. As shown in Figure 3a−f, the broad C1s peak of GO films showed the presence of the basic oxygen-containing functional groups (at 288.6, 287.4, 286.8, 284.7, and 285.7 eV corresponding to C(O)O, C=O, C−O, C (sp^2^) = C (sp^2^) and C–OH respectively.) [29,30]. The quantitative statistics of hydroxyl (-OH) and carboxyl groups (–COOH) as a percentage of the overall number of functional groups are shown in Figure 3g. As depicted in Figure 3a, the pure GO film exhibited 14.8% hydroxyl groups and 14.2% carboxyl groups, which were lower than those in the pure HOGO film (16.1% and 14.4%, respectively), as shown in Figure 3b. This result demonstrates that the HOGO had more oxygen-containing groups at the surface, indicating that more hydrogen bonding can be a incorporated into the layers of GO+HOGO films (15.5, and 14.4%, respectively), as shown in Figure 3c. As shown in Figure 3f, the peak area of hydroxyl groups (15.5 to 14.6%) was reduced when adding borate to the mixture, indicating that tetraborate ions reacted with the hydroxyl groups on the GO surface to form borate ortho-esters (B–O–C) [5,24,28]. The XPS for GO+B and HOGO+B film was measured for further illustration, as shown in Figure 3g,h, which also indicated the reduction of C-OH when borate was added (decreasing from 14.8 to 14.0% and from 16.2 to 14.2%, respectively).

The XPS for GO+B and HOGO+B film was measured for further illustration, as shown in Figure 3d,e, which also indicated the reduction of C–OH when borate was added (decreasing from 14.8 to 14.0% and from 16.2 to 14.2%, respectively). Interestingly, carboxyl groups in the borate-crosslinked GO films were decreased compared to the pure GO films, as shown in Figure 3g, which might have been due to the formation of covalent bonds as a result of further dehydration at 85 °C. The TG-DSC measurements further illustrated the creation of covalent bonds (borate ortho-esters). As shown in Appendix A, the borate-crosslinking GO+HOGO film showed a significant slope change (red) in the thermogravimetric curve at approximately 200 °C, demonstrating the breaking of the borate ortho-esters [24]. For comparison, the GO+HOGO film without adding borate showed a typical GO film thermogravimetric curve. The DTG curves (Appendix A) further confirm covalent bonding, with the GO+HOGO+B sample (red) showing an inverse peak at 200 °C, corresponding to the TGA plateau, suggesting energy absorption for C-O-B bond rupture. The rightward shift of the entire red curve versus the black reference confirms additional energy requirements for C–O–B bond cleavage. The DSC data of borate- crosslinking GO+HOGO (GO+HOGO+B) and GO+HOGO films in Figure 3h further illustrated the enhancement of covalent bonds with the borate addition. The heat flow of the red curve has a significant slope change at approximately 200 °C, compared to the black curve, which depicts that the borate ortho-esters (B–O–C) had strong covalent bonding energy because they absorbed a great deal of heat when broken (approximately 170–210 °C).

### 3.2. WAXS Measurements of the Orientation Enhancement

In addition to illustrating the synergistic enhancement between hydrogen and covalent bonding in our borate-crosslinked GO+HOGO films by XPS and TG-DSC measurements, the orientation of these GO films was improved significantly, as proven by WAXS measurements (the calculation of Herman’s orientation factor ƒ can found in the Characterization section). As shown in Figure 4a,b, the GO and HOGO films showed low Herman’s orientation factors of 0.832 and 0.812, respectively. It should be emphasized that the GO fabricated through the electrochemical method had a large average lateral size of 64.28 µm, while HOGO has a small size of 3.51 µm (Figure 1), resulting in a higher orientation of the GO film than the HOGO film. Since higher orientation is helpful for good mechanical performance, we chose GO as the main framework and HOGO as the additive (2.0, 4.0, 8.0, and 16.0 wt %). As shown in Figure 4c, the GO+HOGO (4 wt %) films showed a much higher Herman’s orientation factor of ƒ = 0.913, indicating that the HOGO generated more hydrogen bonding connections between the layers, which significantly promoted the orientation. Borate addition also improved the orientation of the GO film and HOGO film, as shown in Figure 4d–f. Herman’s orientation factor of the GO+HOGO film was further improved to 0.927 by the borate crosslinking, demonstrating that it can independently improve the orientation of GO film, achieving the best result. Additionally, synergy for orientation enhancement was achieved when both of them were added. For comparison, the orientation of GO+B and HOGO+B films was also tested by WAXS, which showed higher orientation factors (ƒ = 0.901 and, 0.841, respectively) than the pure GO and HOGO films because of the borate bridging, respectively, as shown in Figure 4d,e.

### 3.3. Mechanical Improvement

With the addition of HOGO and borate, the GO films had more interlayer forces and better orientation, and their mechanical properties were anticipated. To investigate the synergistic effect of mechanical properties, various samples were prepared and tested by a CTM 8030 system (see also the Section 2.5), as shown in Figure 4, and tabulated in Appendix A. The typical tensile stress-strain curves of GO, HOGO, and GO+HOGO films were shown in Figure 5a–c and Appendix A. The GO and HOGO films exhibited tensile strength values of 216.5 and 46.5 MPa, moduli of 13.5 and 7.9 GPa, and toughness values of 2.6 MJ m^−3^ and 0.6 MJ m^−3^, respectively. The GO films showed better mechanical properties than the HOGO films, which may have been caused by the difference in lateral size and orientation, as shown in Figure 1 and Figure 4a,b. These mechanical properties are representative and consistent with the previously reported results of GO films [13,29]. With 4 wt % HOGO addition, the tensile strength, modulus, and toughness of the GO+4HOGO film were substantially improved to 338.3 MPa, 22.0 GPa and 6.0 MJ m^−3^, respectively, which are 1.6-, 1.6-, and 2.3-fold higher than those of pure GO film, as shown in Figure 5c and tabulated in Appendix A (other GO+HOGO films). However, when the content of HOGO increased from 8 wt % to 16 wt %, the excessive incorporation of HOGO disrupted the structural framework formed by the large GO nanosheets, leading to a significant deterioration in the mechanical properties.

We also investigated the effects of different borate contents (see also Section 2.4) on the mechanical properties of GO+B, HOGO+B, and GO+HOGO+B films, as shown in Figure 5d–f, Figure 2, Appendix A. When the borate content was increased from 0.03 wt % to 0.12 wt %, the tensile strength and modulus were gradually improved from 205.0 MPa and 29.1 GPa for GO+4HOGO+0.03B to 417.2 MPa and 43.8 GPa for GO+4HOGO+0.12B, respectively, which resulted from the increased covalent bonding bridging, as shown in Figure 5f and tabulated in Appendix A (other GO+HOGO+B films). However, when the borate content was increased to 0.24 wt %, the mechanical properties declined because excess borate could not form covalent bonds within the films. This was equivalent to adding many impurities to the film. For comparison, the GO+B and HOGO+B films were fabricated with the same borate content. The mechanical properties of these films increased or decreased with the borate addition (covalent bonding enhancement), but none of them were higher than those of the GO+HOGO+B films due to the synergistic enhancement by HOGO and borate, as shown in Figure 5d,e, Appendix A [1,5,6,12,13,14,15,16,17,18,19,26,27,28,30,31,34,35,36,37,38,39,40,41,42,43,44].

According to the above results, optimized mechanical properties of the GO film by borate crosslinking were obtained (GO+4HOGO+0.12B) after investigating various HOGO and borate additions. The optimal borate-crosslinked GO+HOGO film had a tensile strength of 417.2 MPa, a toughness of 2.5 MJ m^−3^, and a Young’s modulus of 43.8 GPa, which are 1.6-, 2.2-, and 2.9-fold higher than those of the GO+B film (191.0 MPa, 27.4 GPa, and 0.9 MJ m^−3^) and 2.3-, 3.6-, and 5.06-fold higher than those of the HOGO+B film (116.7 MPa, 18.7 GPa and 0.5 MJ m^−3^), respectively, as shown in Figure 5g. To our knowledge, the GO+HOGO+B film exhibited the best mechanical performance of GO-based films, considering its highest tensile strength with relatively high modulus and toughness (Figure 5h) [1,5,6,12,14,15,16,17,18,19,27,31,33]. Its mechanical properties were also better than those of natural materials [7,8,9,10], and many reduced GO (rGO) films [1,13,31,32,45]. The detailed mechanical properties of these GO films are tabulated in Table 1. Interestingly, in addition to the borate-crosslinked GO+HOGO film, the GO+4HOGO film showed better mechanical properties (338.3 MPa, 22.0 GPa and 6.0 MJ m^−3^) than most of the GO films in the literature [1,5,6,12,13,14,15,16,17,18,19,28,31,32,33,34] and our GO and HOGO films, as shown in Figure 5g. These results indicate that plastic GO films can transfer to the elastic GO films by adding HOGO-induced hydrogen bonding and borate-induced covalent bonding between the layers, which may be suitable for commercial applications. Moreover, if the borate covalent bonding (C–O–B) could be broken apart or recovered by physical or chemical destruction without damage to the intrinsic graphene framework, it might be possible to interconvert plastic and elastic graphene films, offering valuable opportunities for further study.

## 4. Conclusions

In conclusion, we successfully enhanced the mechanical properties of GO films through a synergistic approach with the appropriate addition of both HOGO and borate. XPS and TG-DSC confirmed hydrogen and covalent bonding induced by HOGO and borate, respectively. The borate-crosslinked GO+HOGO film had an optimized orientation factor of 0.927, which is much higher than the values of 0.812–0.832 for pure GO films. It also exhibited exceptional mechanical properties, with an impressive strength (417.2 MPa), high modulus (43.8 GPa), and relatively high toughness (2.5 MJ m^−3^), all of which were significantly higher than those of both the individual GO and HOGO films. Moreover, the borate-crosslinked GO+HOGO had the best mechanical performance for GO-based films, surpassing pure GO films, high-strength and high-modulus GO films in the literature, natural materials, and most of rGO films. This innovative and scalable assembling strategy presents a promising avenue for achieving high strength GO-based films with high modulus and relatively high toughness, thereby enhancing the potential for commercial applications of GO films.

## Figures and Tables

**Figure 1 nanomaterials-15-00630-f001:**
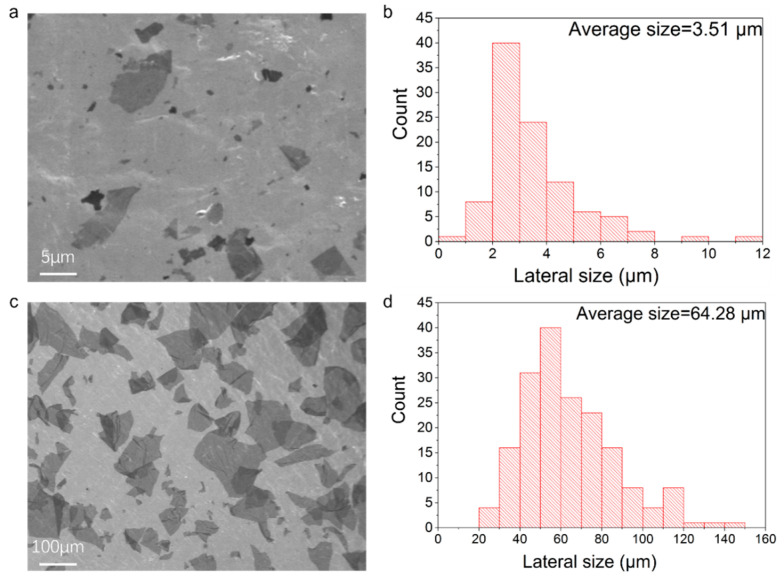
Size characterization of HOGO and GO nanosheets. (**a**,**c**) Scanning electron microscope (SEM) images of HOGO and GO nanosheets, respectively. (**b**,**d**) Width distribution of GO and HOGO nanosheets, indicating that the average size of GO nanosheets is about 3.51 and 64.28 μm.

**Figure 2 nanomaterials-15-00630-f002:**
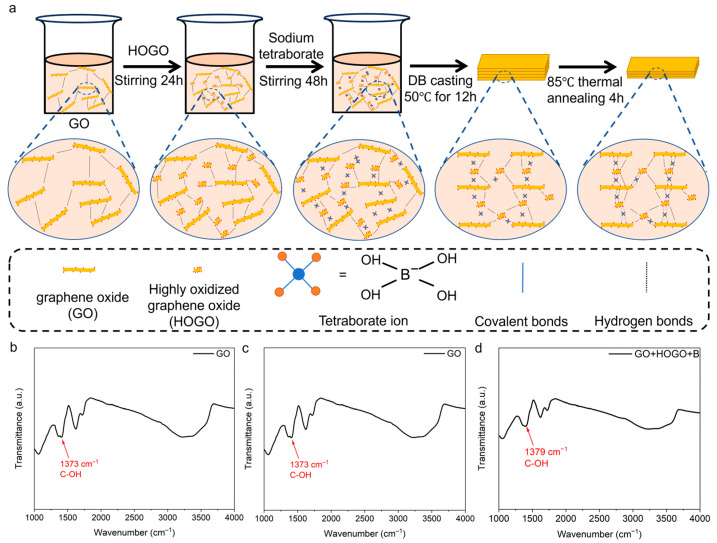
(**a**) Fabrication process of borate-crosslinked GO+HOGO films. (**b**–**d**) Fourier transform infrared (FT-IR) spectroscopy analysis of the fabrication process. See also Appendix A.

**Figure 3 nanomaterials-15-00630-f003:**
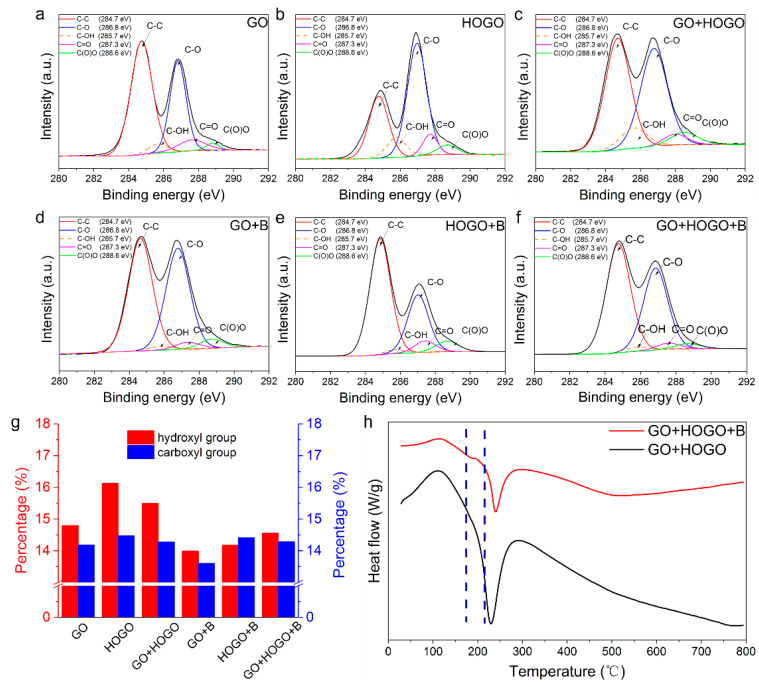
Characterization of the different GO films. (**a**–**f**) X-ray photoelectron spectroscopy (XPS) measurement spectra for carbon in GO, HOGO, GO+HOGO, GO+B, HOGO+B and borate-crosslinked GO+HOGO films, respectively. (**g**) Comparison of the percentage of hydroxyl and carboxyl groups in different GO films. (**h**) Differential scanning calorimetry (DSC) curves of borate-crosslinked GO+HOGO films (red) and GO+HOGO films (black), respectively. See also Appendix A.

**Figure 4 nanomaterials-15-00630-f004:**
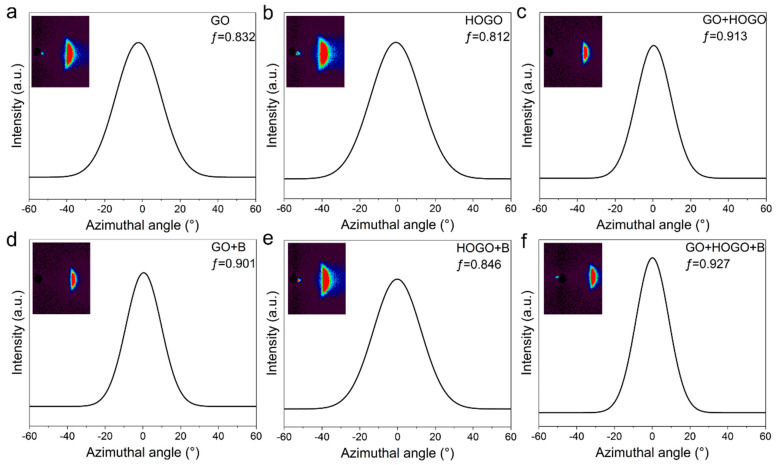
The wide-angle X-ray scattering (WAXS) measurements of the GO and borate-crosslinked GO films. (**a**) GO films. (**b**) HOGO films. (**c**) GO+HOGO films. (**d**) Borate-crosslinked GO films. (**e**) Borate-crosslinked HOGO films. (**f**) Borate-crosslinked GO+HOGO films.

**Figure 5 nanomaterials-15-00630-f005:**
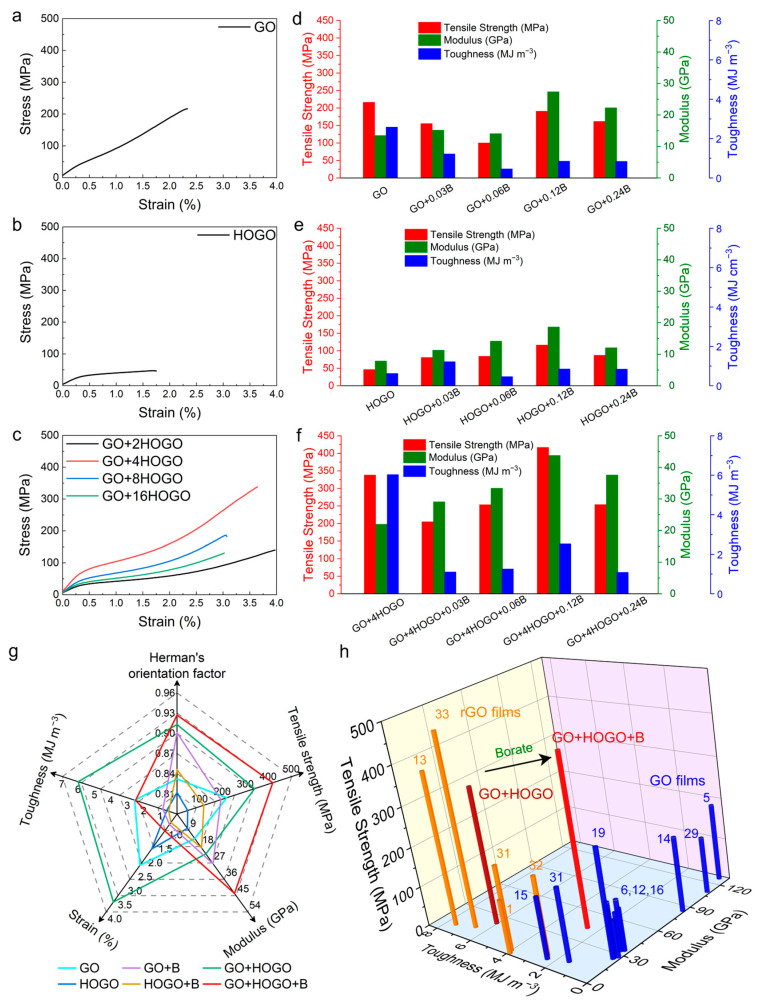
Mechanical properties of GO and borate-crosslinked GO films. (**a**,**b**,**c**) Typical stress strain curves for GO, HOGO and GO+HOGO films. (**d**,**e**,**f**) Comparison of tensile strength (red), modulus (green), toughness (deep blue) for GO, HOGO, GO+HOGO and their borate-crosslinking films, respectively. (**g**) A radial plot comparing the Herman’s orientation factor, tensile strength, modulus, strain and toughness, for typical GO films (azure), HOGO films (deep blue), borate crosslinking GO films (purple), borate crosslinking HOGO films (deep yellow), GO+HOGO films (green), borate crosslinking GO+HOGO films (red). (**h**) Comparison of the tensile strength, modulus, and toughness of our GO+HOGO and borate crosslinking GO+HOGO films with the tensile strength, modulus and toughness of previously reported high mechanical GO (blue) and rGO (orange) films bridged through ionic, covalent and hydrogen bonding [1,5,6,12,13,14,17,18,19,31,32,33]. See also Appendix A and Appendix A.

**Table 1 nanomaterials-15-00630-t001:** Mechanical properties of GO+HOGO, borate-crosslinking GO+HOGO films and other natural materials, GO-based materials, in modulus, tensile strength and strain, including GO films linked by ionic bonding, covalent bonding and synergistic enhancement. More mechanical properties of other graphene-based films can see Appendix A.

	Reference	Materials	Modulus (GPa)	Tensile Strength (MPa)	Toughness (MJ m^−3^)
Natural Materials	[10]	Nacre	26.0–42.0	200.0	2.6
	[10]	Bone	10.0–15.0	50.0–200.0	2.0–10.0
	[9]	Bamboo	20.0–40.0	-	-
GO-based Materials	[12]	GO+Ca^2+^	28.1 ± 1.2	125.8	0.3
		GO+Mg^2+^	27.9 ± 1.8	87.9	0.1
	[45]	GO+Zn^2+^	35.2	142.2	0.3
	[17]	GO+PAA	33.3 ± 2.7	91.9	0.2
	[6]	GO+GA	26.0−34.7	101.0	0.3
	[16]	GO+PGO	35.1 ± 1.7	119.0 ± 27.0	0.4
	[5]	GO+B	127.0 ± 4.0	185.0	0.1
	[28]	GO+B	109.9	135.0	0.1
	[14]	GO+PDA+PEI	84.8 ± 2.9	179.0	0.2
	[19]	GO+PVA	25.3 ± 0.5	255.7	2.5
	[31]	GO+PDA	11.3 ± 4.5	175.0	1.5
	[18]	GO+PVA	40.3	80.2	0.1
		GO+PMMA	10.1	148.3	2.4
	[15]	GO+Al^3+^	26.2 ± 4.6	120.0	0.2
	Our work	LOGO+HOGO	22.0	338.3	6.0
		LOGO+HOGO+B	43.8	417.2	2.5

## Data Availability

All data supporting the findings of this study are included in this published article and its Appendix A.

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
