# Peer review of "Mechanical Improvement of Graphene Oxide Film via the Synergy of Intercalating Highly Oxidized Graphene Oxide and Borate Bridging"

_nanomaterials, 2025, doi:10.3390/nano15080630_

Round 1

Reviewer 1 Report

Comments and Suggestions for Authors

This manuscript reports an interesting study regarding the fabrication of GO-based films with enhanced mechanical properties. Overall, the text is well written and well organized. A few issues should be addressed, as listed below.

  • At the beginning of section 3.1, the text discusses FT-IR characterization, recalling Figure 1. However, Figure 2 should be recalled instead.
  • At the end of page 5, the text reports that from Figure S1, it is possible to infer that “the defects and the layer distance of GO-based films will be reduced by the HOGO nanosheets and Borate adding.” Adding more details about how such a conclusion can be drawn would be useful.
  • The middle upper paragraph on page 7 recalls Figure S1a to refer to TGA results. However, Figure S1a does not report TGA results.
  • At the beginning of Section 3.3, the text recalls Figure 4 instead of Figure 5.
  • The addition of borate increased the strength and stiffness of GO+HOGO films but reduced their toughness. Some comments about this result should be added.
  • On page 10, the mechanical properties of the fabricated films are compared with those characterizing natural materials: “To our knowledge, the GO+HOGO+B film has the best mechanical performance for GO-based films considering its highest tensile strength with relative high modulus and toughness (Figure 5h)[1,5,6,12,14-19,32,36,37], and also has better than those natural materials [7-10]”. However, the reported comparison should be better contextualized because natural materials (the authors should also specify which materials they refer to since the terminology is too generic) feature additional properties other than good mechanical properties.
  • The concluding paragraph of section 3.3 (“These results indicated that the plastic GO films could transfer to the elastic GO films … it might be possible to interconvert plastic and elastic graphene films, which would be valuable for further study.”) is not very clear. A revision is suggested to explain more specifically the main findings and how these can be exploited for future applications.

Author Response

We thank the reviewer for the comments very much. We have provided detailed responses and explanations to the reviewer’s comments.

Reviewer 2 Report

Comments and Suggestions for Authors

Quan, He and Ding have presented an original study into the mechanical improvement of graphene oxide film using a novel combination of borate cross linker plus a highly oxidised GO component.  The manuscript is generally well-presented with good scientific English and I would recommend publication subject to some relatively minor clarifications.  

The methodology used seems to lead to a significant improvement in a range of mechanical properties and the results are backed up with detailed analysis of the structure, thermal and bonding properties by XRD, TGA and XPS respectively.

Suggested corrections:

Abstract: 

...enhancement in the orientation of the resulting GO films...  should be ...enhancement in the orientation of the GO in the resulting films...

In the introduction, there is a good range of literature for other GO materials, but it is not always clear whether the mechanical properties e.g. modulus are measured in the same way.  Is this always tensile modulus, or modulus measured in some other way.  I think some clarification here would help to validate the improvements reported later on.  Similarly there is an open question about how comparable the other results are when for example the flake size or exfoliation may be different.

The authors own results are summarised at the end of the introduction, which is an unusual placement of this material.  A short statement of the aims/strategy/hypothesis might be more  usual.

Figure 1 caption is a little unclear.  I think that the average width of (HOGO and) GO nano sheets is 3.51 and 64.28  um respectively.

2.2. Experimental preparation - it was not clear to me how the dry components (GO minus H2SO4 and the KMnO4)  were mixed or was there some residual sulfuric acid?

2.2. I found the description of formation of s 'single layer of GO' with the shaker a bit unclear.  Is this a single (molecular thick) layer or a layer that is one macroscopic sheet.  If the latter, how thick is it?

Figure 2a. The illustration of the tetraborate ions in the oval diagrams is difficult to make out because they are so small.

Section 3.  Results and Discussion.  

Please clarify the nomenclature at first point of use.  Sodium tetraborate, Borate and B are used interchangeably to mean the same thing.

The statement "To our knowledge sodium...." appears twice.

Near Bottom of page 6, Interestingly the percentage of carboxyl groups... decreased...

At the start of page 7, there is quite a detailed description of the TGA curves that are in the supporting information, but not the main article.  This is a bit confusing for the reader and it would be better to make sure that the figures are included in the main text or the detailed text is put in the supporting information with just the short summary in the main text.

In the same paragraph there is a reference to figure 2h, which does not exist in the main article or the supporting information.

Page 8. The mechanical testing results are (I think) the most significant part of the article.  For readers who are not specialists in this kind of mechanical testing it would be helpful to explain very briefly how each measure is derived from the raw data of the stress-strain curves.

p9 paragraph before table 1.  'when the HOGO concentration was increased from 8 wt% to 16 wt %...' might be clearer than the text as written.

table 1.  It is not clear to me why the results from reference 25 are not also included here.  These results seem to be comparable to the results achieved in this study and merit being included.  Given the similarity of their approach I think it may be valuable to conclude that that a general approach to reinforcement that utilises more than one strategy simultaneously is the key to simultaneously improving multiple measures of modulus, toughness etc.

page 10. I think that reinforcement of GO is known to become poorer at excessive crosslinker concentration.

page 11 Data availability statement is a bit weak.  Data are not easily available as presented.

Author Response

We sincerely appreciate the reviewer's positive comments and valuable recognition of our work. We are grateful for the reviewer's constructive suggestion and corresponding revisions have been made as detailed below.
